# *Lavandula stoechas* subsp. *luisieri* (Rozeira) Rozeira: Variability of Chemical Composition of Essential Oil in Wild Populations

**DOI:** 10.3390/plants14223435

**Published:** 2025-11-10

**Authors:** Francisco Márquez-García, David García-Alonso, María del Carmen García-Custodio, Francisco María Vázquez-Pardo

**Affiliations:** Centro de Investigaciones Científicas y Tecnológicas de Extremadura (CICYTEX), Instituto de Investigaciones Agrarias Finca La Orden-Valdesequera, Área de Biodiversidad Vegetal Agraria, Autovía A-5 Km 372, 06187 Guadajira, Badajoz, Spainfrvazquez50@hotmail.com (F.M.V.-P.)

**Keywords:** essential oil, *Lavandula stoechas* subsp. *luisieri*, necrodol derivatives, spanish lavender

## Abstract

*Lavandula stoechas* subsp. *luisieri* (Rozeira) Rozeira is an aromatic shrub endemic to the south-west of the Iberian Peninsula. It is distinguished by being the only species of lavender that contains necrodol derivatives in its essential oil. This study aims to evaluate the diversity of the chemical composition of *L. stoechas* subsp. *luisieri* essential oil and how it differs from essential oils of other lavender species with which it shares its habitat and with which it can hybridize. The variability in the chemical composition of *L. stoechas* subsp. *luisieri* essential oil has been evaluated across 66 populations distributed among 14 areas in southwestern Iberian Peninsula. The main compounds present in the essential oil of *L. stoechas* subsp. *luisieri* are *trans*-α-necrodyl acetate (20.68 ± 4.17%), 1,8-cineole (7.79 ± 7.14%) and *trans*-α-necrodol (8.66 ± 2.18%). Other compounds may occasionally be present in percentages greater than 6.00%, such as α-cadinol, linalool, lavandulyl acetate, fenchone and camphor. Two essential oil types have been identified in the essential oil of *L. stoechas* subsp. *luisieri*: (1) *trans*-α-necrodyl acetate-1,8-cineole, and (2) *trans*-α-necrodyl acetate, with little or no presence of 1,8-cineole. Furthermore, the absence or very low percentage of camphor (0.16–7.61%) and fenchone (0.00–7.39%) has been confirmed as a unique characteristic of this essential oil. The obtained results provide clear differentiation of the essential oil of *L. stoechas* subsp. *luisieri*, thus enabling more accurate studies to be carried out on its bioactive properties, which are of great interest to industry.

## 1. Introduction

The genus *Lavandula* L. (Lamiaceae) comprises approximately 40 species, as well as a variety of infraspecific taxa and hybrids (around 75 taxa). These species are, generally, endemics (with a narrow distribution) that are primarily distributed in the Mediterranean region, encompassing from north Atlantic Islands to South Asia. There are important hotspots in Canary Islands (5 species and 7 subspecies), Morocco (9 species, 4 of those are endemic species) and South of the Arabian Peninsula and West of the Horn of Africa (18 species, 12 of them endemic) [1,2].

Several lavender species have been used by humans since ancient times. First written references about medicinal use of lavender were from first century BC, in “De Materia Medica (pp. 460–461)” from Dioscorides, where is mentioned the *stoechas* term refer to the use of lavender species attributable to *Lavandula stoechas* L. and *Lavandula vera* DC. (=*Lavandula angustifolia* L.). Their uses extend from Egyptian, Roman, Greek and Persian civilizations [3,4,5]. The wide use of these species as aromatic and medicinal plants has been maintained over millennia, thanks to their recognized properties (sedatives, carminatives, antidepressants and anti-inflammatory), and, during First World War, after the discovery of the antibiotic capacity of the *L. angustifolia* essential oil, the growth of its essential oil demand contributed to the expansion of lavender crops [3,6,7]. At present, the cultivation of lavender is predominantly focused on three species, *L. angustifolia* (true lavender), *L. latifolia* Medik. (spike lavender), and their hybrid *L. × intermedia* Emeric *ex* Loisel. (lavandin), which are present over several countries, among which stand out: Bulgaria, France, United Kingdom, Spain, Italy, Russia, Ukraine, Moldova, Romania, Hungary, Poland, Turkey, Morocco, United States, Australia, South Africa, and China [8,9,10,11,12].

In recent decades, a significant number of research projects have been conducted on the chemical composition, traditional uses and medicinal properties of essential oils from other lavender species, being especially high the number of published studies about *L. stoechas* [13,14].

Two subspecies of *L. stoechas* are recognized: *L. stoechas* subsp. *stoechas* and L. *stoechas* subsp. *luisieri* (Rozeira) Rozeira (Figure 1), which present differences both morphologically, in the chemical composition of their essential oils, and geographic distribution.

*L. stoechas* subsp. *stoechas* (French lavender) is an aromatic shrub native to the coastal areas of the Mediterranean region, penetrating inland where topography allows and at altitude where the climate is suitable, on coastal and internal mountain ranges, at altitudes between sea level and 1000(-1350) m (Figure 2a) [1].

*Lavandula stoechas* subsp. *luisieri* (Spanish lavender) is an endemic shrub to the SW of the Iberian Peninsula. It inhabits a variety of environments, including cliffs and coastal dunes, as well as inland mountain ranges, at altitudes below 900–950 m above sea level (m.a.s.l.) (Figure 2b) It is associated with pioneer plant formations of Mediterranean scrublands and with the shrub stratum of the understory in more or less dense forests of oaks (*Quercus* L. spp.) and pines (*Pinus* L. spp.) [1,15,16].

**Figure 2 plants-14-03435-f002:**
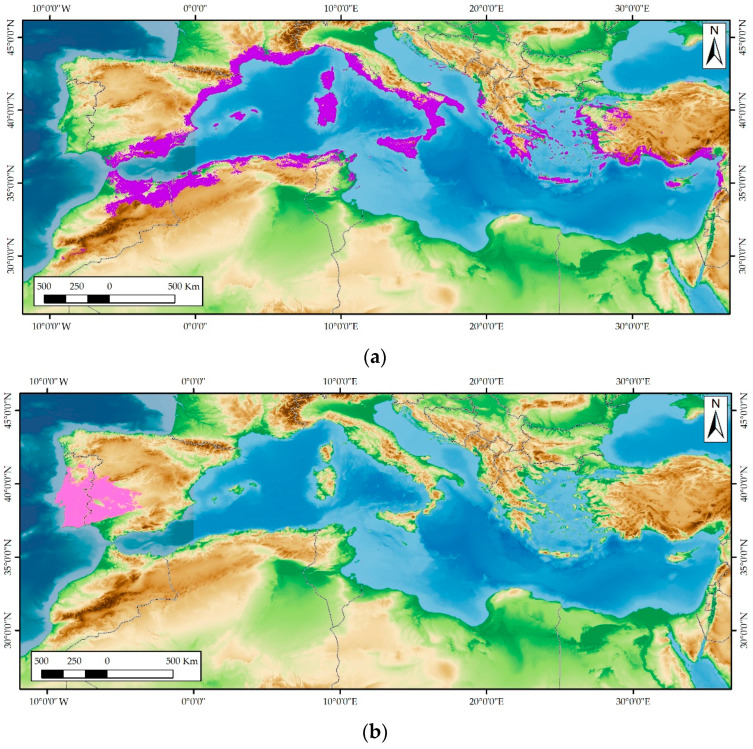
Distribution maps of *L. stoechas* subspecies: (**a**) *L. stoechas* subsp. *stoechas* (purple color); (**b**) *L. stoechas* subsp. *luisieri* (pink color) (Source: study of herbarium vouchers. Unpublished work).

(a) Morphological differences: *L. stoechas* subsp. *stoechas* is characterized by ascending stems up to 70 cm high. The inflorescence peduncle is up to 2(3) cm long and no longer than the spike itself. The indumentum of the fertile bracts is woolly (scattered to dense) with simple hairs. *L. stoechas* subsp. *luisieri* has erected stems up to 150(200) cm in height, an inflorescence peduncle reaching up to 6 cm, and the indumentum of the fertile bracts velvety, denser on the venation, with short and branched hairs [1,15,16].

(b) Chemical composition of *L. stoechas* subsp. *stoechas* essential oil has as main components fenchone [17,18,19,20,21,22,23,24,25,26,27,28,29,30,31,32,33,34], camphor [35,36,37,38,39,40,41,42,43,44,45], and 1,8-cineole [46], and rarely other majority components as linaool [47,48] and pulegone [49]. Instead, *L. stoechas* subsp. *luisieri* has an essential oil rich in necrodol derivatives, mainly *trans*-α-necrodyl acetate, accompanied by other components such as 1,8-cineole, lavandulyl acetate, lavandulol, viridiflorol, fenchone, and camphor [50]. However, these essential oils do not show a stable chemical composition in their whole natural distribution area and wide ranges of variation in their major components can be observed [13,50].

*L. stoechas* subsp. *luisieri* essential oil differs from the rest of lavender essential oils because it is the only one that contains necrodol derivatives (*trans*-α-necrodyl acetate, *cis*-α-necrodyl acetate, *trans*-α-necrodol, and *cis*-α-necrodol) [13,50]. First studies developed about chemical composition over this essential oil indicated that 1,8-cineole (0.10–43.20%) and *trans*-α-necrodyl acetate (6.30–28.40%) are the main components and camphor and fenchone have very low percentages of presence (0.40–6.30% and 0.00–5.40%, respectively) [51,52,53]. Subsequent studies on the potential applications of the essential oil revealed the presence of camphor as the primary component (42.90–60.30%), with *trans*-α-necrodyl acetate having a residual presence (0.00–12.39%) [54,55,56]; and in other research, the majority component has been identified as fenchone, with a percentage of up to 36.77% [56,57].

The presence of *trans*-α-necrodyl acetate and other necrodol derivatives renders the *L. stoechas* subsp. *luisieri* essential oil particularly distinctive, thereby evoking the interest of the cosmetic and pharmaceutical industries. Several studies give to it good antifungal and antibacterial activity [58,59,60,61,62,63], moderate antioxidant capacity [55,63,64,65], good anti-inflammatory activity [66,67,68,69] and high inhibition capacity of proBACE-1 (enzyme related to Alzheimer disease) [69], good effects over trypanosomatid diseases [56,70], and over infections made by *Leishmania* species (Protozoal) [71]. Finally, its antifeedant activity is subject to variation, exhibiting low activity against *Hyalomma lusitanicum* [56], and between low and high against *Spodoptera littoralis*, *Leptinotarsa decemlineata*, *Myzus persicae*, and *Rhopalosiphum padi*, among others [58,72,73,74].

The divergent properties exhibited by this species (e.g., anti-inflammatory and antioxidant properties) in conjunction with the diversity observed in the chemical composition of its essential oil, the morphological similarity with *L. stoechas* subsp. *stoechas*, and the high hybridization capacity of *L. stoechas* subsp. *luisieri* with other lavender species (*L. pedunculata* (Mill.) Cav. and *L. viridis* L’Hér.) necessitate a comprehensive evaluation of the morphological variability and chemical composition of its essential oil throughout its natural distribution area. The objective of the present study is twofold: firstly, to ascertain the presence or absence of diverse chemotypes in *L. stoechas* subsp. *luisieri*, and secondly, to identify the chemical components that contribute to its economic value.

## 2. Results

A *total* of 66 populations of *L. stoechas* subsp. *luisieri* were analyzed. The distribution of these populations is primarily concentrated in inland mountainous regions, with elevations ranging up to 912 m a. s. l., and they are seldom observed in coastal areas or alluvial plains, with elevations ranging from 36 to 192 m a. s. l. (see Figure 3, Table 1 and Appendix A).

### 2.1. Essential Oil Yield

The mean essential oil (EO) yield of the *L. stoechas* subsp. *luisieri* populations obtained is 3.08 ± 1.42 g/kg (grams of EO/kg fresh sample), with the lowest yield (1.08 g/kg) and highest yield (8.17 g/kg) located in Serra de Alvelos (Portugal) and Montes de Toledo (Spain), respectively. With regard to the average essential oil yield, the majority of the areas studied demonstrate an average yield of 2.50–3.50 g/kg. The lowest yields were observed in the Portuguese mountain ranges of Serra de Alvelos and Serra da Gardunha, with average values of 1.46 ± 0.45 g/kg and 2.23 ± 0.60 g/kg, respectively. The lowest yield was recorded in the isolated population located in the alluvial plain of the Guadiana River (Vegas Bajas), with a value of 1.75 g/kg. The areas with the highest average yields are the Montes de Toledo Mountain range (Spain) and the Guadiana River floodplain as it passes through Portugal (Guadiana International), with values of 5.20 ± 1.30 g/kg and 4.25 ± 1.07, respectively (Table 2 and Appendix A).

### 2.2. Chemical Composition

The chemical composition of *L. stoechas* subsp. *luisieri* essential oil is highly variable. A total of 159 distinct compounds were detected in the 66 samples analyzed, and their presence exhibited significant variability (Appendix A). The chemical composition of the essential oil is predominantly comprising between 80–89, 90–99, and 70–79 different compounds, in 26, 21, and 12 samples, respectively. Only two samples (belonging to populations 17 and 18) have fewer than 70 different compounds in the essential oil, and four samples have more than 100 compounds (101 compounds in population 4, and 100 compounds in populations 47, 54, and 63). Furthermore, this essential oil is characterized by a high number of unknown compounds (20 in total), some of which may account for up to 3.60–3.71% of the total composition, as well as 4–20% of unidentified compounds. The percentage of compounds identified in each essential oil sample ranges from 80.99% (population 1) to 96.70% (population 19), and the average percentage of compounds identified in the essential oil of the 66 populations studied is 88.19 ± 3.09%.

A total of 24 different chemical compounds were identified among the ten compounds with the highest percentage of presence in at least one essential oil sample (Appendix A and Table 3). Of these, *trans*-α-necrodyl acetate, *trans*-α-necrodol, and lavandulyl acetate are present in all samples among the 10 major compounds. The following compounds were detected in all essential oil samples analyzed: 1,8-cineole, camphor, linalool, 5-methylene-2,3,4,4-tetrame-2-cyclopentenone, *cis*-α-necrodol, α-pinene, arbozol, eudesma-3,7(11)-diene, and *cis*-α-necrodyl acetate; however, these compounds did not always appear among the top ten. It is noteworthy that the presence of fenchone, cymene isomer, viridiflorol, α-cadinol, copaborneol and several unidentified components (monoterpene, sesquiterpene or ester) may be observed among the ten compounds, although they may not be present in the essential oil.

*Trans*-α-Necrodyl acetate was identified as the predominant compound in 60 samples and the secondary compound in the remaining 6 samples analyzed, with an average percentage of 20.68 ± 4.17%. 1,8-Cineole was the predominant compound in six samples and the second most abundant in twenty samples. The mean percentage of presence was found to be 7.79 ± 7.14%, with a wide range of variation observed, from traces (<0.05%) to 24.24%. *Trans*-α-necrodol is the second most abundant compound, appearing in 37 samples, with an average percentage of presence of 8.66 ± 2.18%.

As illustrated in Figure 4 and outlined in the Appendix A, the distribution of the eight primary components in the essential oil of *L. stoechas* subsp. *luisieri* is depicted by geographical area. The populations studied in the Spanish mountain ranges of Alconera, Villuercas, and Sierra Morena de Sevilla, generally, have an essential oil that is rich in *trans*-α-necrodyl acetate and 1,8-cineole, with 1,8-cineole being the main component in some cases. In other areas, the percentage of *trans*-α-necrodyl acetate generally exceeds that of 1,8-cineole. This is of particular significance in the populations studied in the Serra da Gardunha, Serra de Alvelos, Serra de São Mamede, and Sierra de San Pedro Mountain ranges, and in the alluvial zone population of Vegas Bajas, where 1,8-cineole has very low percentages of presence. The mean percentage of presence in these areas ranges from 0.62 ± 0.81% in Serra de São Mamede to 2.54 ± 1.47% in Serra da Gardunha.

In contrast, the *trans*-α-necrodol exhibited a certain degree of stability, with levels ranging from 5–12% across all areas. Of the other compounds, the presence of α-cadinol is worthy of note in the populations of the Montes de Toledo, Sierra Morena de Sevilla, Sierra de San Pedro y Villuercas, and Fenchone mountain ranges, principally in Serra de Monchique, where it can reach percentages of up to 12.76%.

### 2.3. Cluster Analysis

Hierarchical clustering was utilized to investigate the correlation between the chemical composition of the 66 populations of *L. stoechas* subsp. *luisieri* (Figure 5). The dendrogram obtained reveals the presence of two clusters, which are further subdivided into four subclusters at various hierarchical levels.

The two primary groups (clusters I and II) of the dendrogram demonstrate a clear delineation of the essential oil samples according to their chemical composition (see Table 4). The primary distinctions between these groups pertain to the percentage of 1,8-cineole present, which ranges from 0.04% to 12.84% in cluster I and from 10.50% to 24.24% in cluster II, and the presence or absence of other minor compounds, including copaborneol, α-cadinol, and several unknown compounds.

Cluster I is further subdivided into four subclusters, which are distributed across a range of hierarchical levels. The initial subdivision successfully isolates a sample (subcluster Ib) that is distinctly different from the others due to its elevated camphor content (7.61%), which is the highest percentage detected in all the samples analyzed in this study. Subcluster Ia is further subdivided into three groups (Ia1′, Ia1″, and Ia2) that are primarily categorized based on the percentages of *trans*-α-necrodyl acetate and 1,8-cineole present. However, there is overlap between the ranges of values due to the influence of other minor compounds that are not among the top ten (e.g., α-cadinol, camphor, fenchone).

Cluster II is further subdivided into two subclusters (IIa and IIb). Subcluster IIb is distinguished by its unique composition, with two samples of essential oil exhibiting a high concentration of fenchone, approximately 13% of the total composition. The remaining three groups contained within subcluster IIa are distinguished by a high percentage of 1,8-cineole (greater than 10%) and its relationship with the percentage of *trans*-α-necrodyl acetate. In group IIa2, the range of 1,8-cineole is from 10.50% to 14.01%, and for *trans*-α-necrodyl acetate, it is from 18.17% to 26.04%. In addition, α-cadinol is present in all samples, with a range of 2.99% to 10.78%. Group IIa is subdivided into two subgroups, designated IIa1′ and IIa1″. Subgroup IIa1′ consists of samples in which the percentage of *trans*-α-necrodyl acetate (16.59–20.46%) exceeds that of 1,8-cineole (12.42–18.60%), while subgroup IIa1″ comprises samples in which the percentage of 1,8-cineole (19.84–24.24%) surpasses that of *trans*-α-necrodyl acetate (11.11–19.48%).

Regarding the relationship between the chemical composition of the essential oil and the harvesting area, Figure 5 shows that there is a weak relationship between samples from the same population, except for those from Montes de Toledo, where five of the ten populations are included in subgroup IIa2 (cluster II).

## 3. Discussion

### 3.1. Geographical Distribution

The populations of *L. stoechas* subsp. *luisieri* collected are predominantly located in mountainous areas (54 populations), with sporadic occurrences in more or less flat areas associated with the course of major rivers (Guadiana and Tagus rivers) (5 populations) and in areas of the Portuguese Atlantic coast (7 populations). These populations, in conjunction with those from other studies [51,53,57,58,63,72], indicate a clear preference of the species for mountainous areas in the interior of the Iberian Peninsula (Figure 6a).

### 3.2. Essential Oil Yield

The essential oil yield obtained in this study (Table 1) ranges from 1.08 g/kg (0.11% *w*/*w*) to 8.14 g/kg (0.81% *w*/*w*), with an average yield of 3.44 ± 0.87 g/kg (0.34 ± 0.01%). In the harvests from Almadén de la Plata, located in the north of Seville and included in the study area of Sierra Morena de Sevilla, yields of 0.20% and 0.30% *w*/*w* fresh weight [53,75] were obtained. These values are comparable to, or slightly lower than, those obtained in the eight populations analyzed in the Sierra Morena de Sevilla area (average yield: 0.31 ± 0.07% *w*/*w*. As demonstrated in the various harvests conducted in central Portugal (Penamacor, Vila Velha de Ródão, Mata, and Casal da Fraga), the yields ranged from 0.07% to 0.38% *w*/*w* [57], which is analogous to the yields obtained for the mountain ranges of central Portugal studied, Serra da Gardunha and Serra de Alvelos, where the minimum and maximum values recorded were 0.11% and 0.30%, respectively (Figure 6b). Other studies indicate higher yields of up to 2.80%; however, these units are expressed in terms of dry matter volume/weight (% *v*/*w*) [51,54], and are therefore not comparable with the data obtained in our study.

### 3.3. Chemical Composition of Essential Oil

The chemical composition of the essential oil of *L. stoechas* subsp. *luisieri* is distinguished by the presence of necrodol derivatives, which are absent in other lavender species. Preliminary studies undertaken by García-Vallejo [51,52] utilizing 37 specimens from 27 locations have indicated that *trans*-α-necrodyl acetate (6.30–28.80%) is the predominant compound in 24 essential oil samples, while 1,8-cineole (0.10–43.20%) is the primary compound in the remaining 13 samples. Conversely, camphor (0.40–6.40%) and fenchone (0.00–5.40%), which are major compounds found in other morphologically similar lavender species (*L. stoechas* subsp. *stoechas*, *L. pedunculata*, and *L. viridis*), are present in low concentrations. Subsequently, the characteristic profile of the essential oil of *L. stoechas* subsp. *luisieri* was confirmed in samples collected in Almadén de la Plata (in the north of Seville) [53,76], and novel unidentified compounds began to appear, such as: The following compounds have been identified: 1,1,2,3-tetramethyl-4-hydroxymethyl-2-cyclopentene, 2,3,4,5-tetramethyl-2-cyclopenten-1-one, and 2,3,5,5-tetramethyl-4-methylene-2-cyclopenten-1-one (=5-Methylene-2,3,4,4-tetramethylcyclopent-2-enone).

The major compound in essential oil samples obtained from leaves collected in populations in northern Seville and southern Toledo (with values reaching up to 38.30%) is 2,3,5,5-tetramethyl-4-methylene-2-cyclopenten-1-one [72]. The study also highlights the presence of camphor (with percentages of 51.80% and 53.70% for essential oil samples obtained from flowers and leaves of the same individual), fenchone (with values reaching 22.0% in some samples), 1,8-cineole (with a range of presence between 0.40% and 20.60%, being the major component in three samples), and viridiflorol (with 12.10% being the major compound in a sample of essential oil obtained from flowers). In contrast, necrodol derivatives exhibit remarkably low percentages, particularly *trans*-α-necrodyl acetate, which displays an occurrence percentage of 1.10–3.70% [72].

In the course of further research conducted on samples collected in various areas of Portugal, 1,8-cineole was found to be the main component (25.70–34.30%), followed by *trans*-α-necrodyl acetate (11.30–17.50%) in three areas of the Alentejo coast. Montenegro (Faro), Salir (Loulé), and Vila Real de Santo António [63]. Conversely, in four towns in the south-east of the Beira Interior region (Vila Velha de Ródão, Mata, Penamacor, and Casal da Fraga), *trans*-α-necrodyl acetate was identified as the predominant compound in essential oil samples from leaves (25.23–49.95%) and flowers (9.40–26.99%) in three localities (Penamacor, Mata, and Casal da Fraga). Meanwhile, 1,8-cineole was detected in trace amounts (0.00–4.95%). In the sample from Vila Velha de Ródão, fenchone was identified as the predominant compound, accounting for 19.16% and 36.77% of the essential oil derived from leaves and flowers, respectively. *Trans*-α-necrodyl acetate was the second most abundant compound, contributing 19.38% and 10.27% to the essential oil extracted from leaves and flowers, respectively [57]. The Penamacor population is subsequently used in other studies, in which, together with other properties of the essential oil, the variation in chemical composition between leaves and flowers during the different phenological stages of the plant is evaluated. The research findings indicate that the predominant compound in the sample is *trans*-α-necrodyl acetate (1.80–26.90 per cent), with 1,8-cineole present in substantial concentrations in the leaves (13.90–16.40%), and camphor reaching 42.90% in an essential oil sample obtained from the leaves at the end of the flowering period [54,59]. The most recent populations the focus of study in Portugal are Piodão (in the central region) and Cabo São Vicente (in the Algarve), which exhibit markedly divergent chemical compositions. In Piodão, the essential oil is characterized by a high concentration of *trans*-α-necrodyl acetate (16.00–17.40%), lavandulyl acetate (6.10–7.60%), and *trans*-α-necrodol (7.10–8.40%), while in Cabo São Vicente, the predominant compounds are 1,8-cineole (33.90%), fenchone (18.40%), and *trans*-α-necrodol (4.50%), with *trans*-α-necrodyl acetate exhibiting a significantly lower concentration (3.20%) [58,68,69,71]

Finally, the final population studied is in Montes de Toledo (Pueblo Nuevo de Bullarque), and its essential oil has a chemical composition that is rich in camphor (74.40%), with no traces of the various necrodol derivatives and 1,8-cineole [73].

### 3.4. Principal Compounds of Essential Oil

The observed percentage variation in the compounds that constitute the essential oil of *L. stoechas* subsp. *luisieri* is largely consistent with the results obtained in this study [50,51]. *Trans*-α-necrodyl acetate is typically the most prevalent compound, with concentrations reaching up to 48.22–49.45% in essential oils derived from fresh leaves collected in Mata and Penamacor, respectively [57]. These values exceed those observed in this study, where the maximum concentration of *trans*-α-necrodyl acetate is 30.50% in an essential oil derived from fresh whole plants from a population in Sierra de Aracena and Picos de Aroche. This discrepancy in values can be attributed to the nature of the sample type employed, given that *trans*-α-necrodyl acetate is found in higher concentrations in leaves than in flowers [57].

1,8-Cineole has been found to be present in a wide range of proportions in the essential oil of *L. stoechas* subsp. *luisieri*, ranging from very low percentages to as high as 43.20% in a population from Aznalcollar [51] and 24.24% for a population studied in this work from Sierra Morena de Sevilla. In the present study, the geographical distribution of the samples examined did not appear to be related to the percentage of 1,8-cineole present in the essential oil of *L. stoechas* subsp. *luisieri* (Figure 7). Samples exhibiting high or low percentages of 1,8-cineole may be observed in the same area, as has been previously reported in other studies [51,72].

Other compounds are characteristic of the essential oil of *L. stoechas* subsp. *luisieri*, including camphor and fenchone. The prevalence of both is low (e.g., less than 8.00%), and in some cases, they may even be absent [51,52]. However, in this study, fenchone exceeds 10.00% in two populations (ID numbers 5 and 34, from the Atlantic coast and Serra de Monchique areas, respectively) that appear isolated in the cluster analysis (Figure 5). In addition, other studies have identified samples exhibiting even higher values, including two samples from southern Toledo with values reaching up to 22.00% [72], a sample from Cabo São Vicente (18.20%), and a sample from Vila Velha de Ródão with values up to 36.77% [57]. Morphological analysis (unpublished data) of the populations collected in this study indicates the existence of several individuals with morphologies intermediate between *L. stoechas* subsp. *luisieri* and *L. pedunculata* (hybrids), which may be responsible for the increase in fenchone in the essential oil. A comparable phenomenon has been observed in the sample from Vila Vela de Ródão. The three samples under scrutiny have been found to contain necrodol derivatives, with concentrations ranging from 10.00 to 25.00%. Conversely, in samples from southern Toledo and Cape São Vicente, the presence of necrodol derivatives is found in minimal percentages (<2.00%), thus prompting uncertainty regarding the authenticity of these samples as *L. stoechas* subsp. *luisieri*.

A comprehensive analysis of the extant literature pertaining to the essential oil of *L. stoechas* subsp. *luisieri* reveals the presence of two samples that are characterized by a high concentration of camphor: one from southern Toledo (with 51.80–53.70%) [72], and one from Pueblo Nuevo de Bullarque (with 74.40%) [73]. In both cases, the necrodol derivatives characteristic of the essential oil of *L. stoechas* subsp. *luisieri* are not present, so it is possible that these samples correspond to another species of *Lavandula*, possibly *L. pedunculata* subsp. *sampaioana* (Rozeira) Franco, a species from which samples of essential oil rich in camphor (up to 84.40%) have been found in various locations in the Montes de Toledo and Sierra Morena Mountain ranges [51].

## 4. Materials and Methods

### 4.1. Collection of Plant Materials

In spring 2024 (April–May), 66 populations of *L. stoechas* subsp. *luisieri* were collected, distributed throughout much of the natural distribution area (Appendix A). During fieldwork, the following diagnostic characteristics were utilised to accurately identify populations of *L. stoechas* subsp. *luisieri*: the length of the inflorescence peduncle, the type of indumentum, and the plant size. Subsequently, between five and ten individuals from each population were selected for the purpose of confirming identification. These materials are preserved as vouchers specimens in the HSS Herbarium (Centro de Investigaciones Científicas y Tecnológicas de Extremadura, Lobón, Spain).

In each population (sample), 20 individuals were selected that were representative of the morphological diversity of the species existing in the population. For each individual (subsample), a fraction of the aerial part (stems, leaves, and flowers) was collected individually and stored in raffia bags at room temperature (20–25 °C) for 12–24 h. Next, in the laboratory, each subsample was cut into small pieces, and 25 g of sample were selected at random. Finally, the 20 subsamples or individuals were mixed to obtain the final sample from each population (25 × 20 = 500 g fresh weight). All samples were frozen at −40 °C until distillation.

### 4.2. Essential Oil Extraction

For the extraction of essential oil, each sample (500 g of fresh, frozen material) was placed in a 5000 mL spherical flask and mixed with distilled water (temperature: 20–25 °C) in a proportion of 20% (*w*/*w*). The samples were left to thaw for 10–15 min and then distilled.

The distillation method used was hydro-distillation in a Clevenger-type apparatus for 2 h. The extracted essential oil was stored in an amber vial at 4 °C.

### 4.3. Chemical Characterization of Essential Oils

The chemical analysis of the essential oils was carried out using a combination of two gas chromatography techniques (GC-FID + GC-MS), chemical compounds were identified by CG-MS and quantified by CG-FID. The analysis was performed on Agilent 8890 GC (Agilent Technologies, Inc., Santa Clara, CA, USA) paired with the Agilent 5977B MSD (Mass Selective Detector). Polar column DB-WAX UI (60 m long, 0.25 mm diameter and 0.5 µm film thicknesses) was employed using Helium carrier gas at constant flow of 2 mL/min. Apolar column HP-5MS UI (60 m long, 0.25 mm diameter and 0.25 µm film thicknesses) was employed using Helium carrier gas at constant flow of 1 mL/min. The column temperature started at 50 °C and increased to 240 °C (polar column) and 285 °C (apolar column).

### 4.4. Statistical Analysis

Statistical parameters were calculated using SPSS statistics version 23.0 (IBM, New York, NY, USA) and for the graphical representations of data, MS Excel version 2507 (Microsoft 365 for business, Microsoft, Redmond, Washington, DC, USA).

The percentage composition of the isolated essential oils was used to determine the relationship between the different samples by cluster analysis using the DARrwin, version 6.0.021 (CIRAD-BIOS UMR AGAP, Montpellier, France) [76,77]. For cluster analysis, the Euclidean distance metric was selected as a measure of similarity among samples and the Unweighted Pair Group Method using Arithmetic Averages (UPGMA) was used for cluster definition.

## 5. Conclusions

The essential oil of *L. stoechas* subsp. *luisieri* is clearly distinguishable from any other lavender essential oil due to the presence of necrodol derivatives. These compounds can be found in four forms: *trans*-α-necrodyl acetate, *cis*-α-necrodyl acetate, *trans*-α-necrodol, and *cis*-α-necrodol. *Trans*-α-necrodyl acetate is typically the most prevalent compound in the essential oil. Necrodol derivatives have been shown to possess antifeedant, antioxidant, and antimicrobial properties, which renders them of particular interest to the agrochemical and food industries. In the agrochemical industry, the essential oil of *L. stoechas* subsp. *luisieri*, which is rich in necrodol derivatives, has been found to be effective in combating insect pests. In the context of the food industry, the properties of the aforementioned substances, which include antioxidant and antimicrobial properties, make them suitable for use as preservatives.

Conversely, the presence of necrodol derivatives confers a distinct scent to the essential oil of *L. stoechas* subsp. *luisieri*, differentiating it from other varieties of lavender. This facilitates its utilisation in the production of new fragrances or aromas by the fragrance industry.

The presence of 1,8-cineole has been detected in percentages ranging from trace amounts (<0.10%) to over 25.00%, indicating a considerable degree of variability. This behavior is instrumental in differentiating two types in the essential oil of *L. stoechas* subsp. *luisieri*: one essential oil type is characterized by a high concentration of *trans*-α-necrodyl acetate and a low representation of 1,8-cineole (less than 6%), while the other essential oil type is distinguished by a high concentration of both compounds, with *trans*-α-necrodyl acetate and 1,8-cineole at percentages between 15.00–25.00%, rarely higher, and with 1,8-cineole as the predominant component in some cases.

The presence of 1,8-cineole, linalool, viridiflorol, and lavandulyl acetate, among other compounds, gives the essential oil of *L. stoechas* subsp. *luisieri* anti-inflammatory, analgesic, and antibacterial properties. These properties are of significant value to the pharmaceutical and cosmetics industries in the development of new formulations for creams and lotions.

A further distinguishing characteristic of *L. stoechas* subsp. *luisieri* essential oil is the comparatively low representation of camphor and fenchone, with percentages of presence of less than 8.00%. This characteristic differentiates the essential oil of this species from other species in the *stoechas* section of the *Lavandula* genus (*L. stoechas* subsp. *stoechas*, *L. pedunculata*, *L. viridis*, and their hybrids).

## Figures and Tables

**Figure 1 plants-14-03435-f001:**
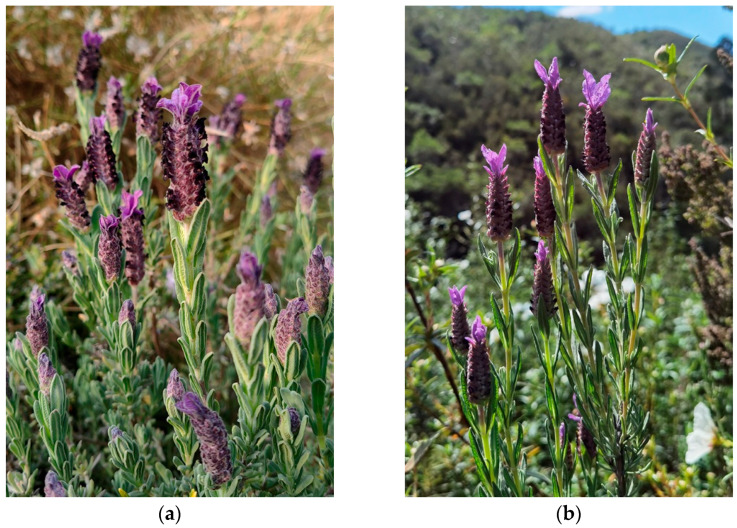
Photography of *L. stoechas* subspecies: (**a**) *L. stoechas* subsp. *stoechas*; (**b**) *L. stoechas* subsp. *luisieri*.

**Figure 3 plants-14-03435-f003:**
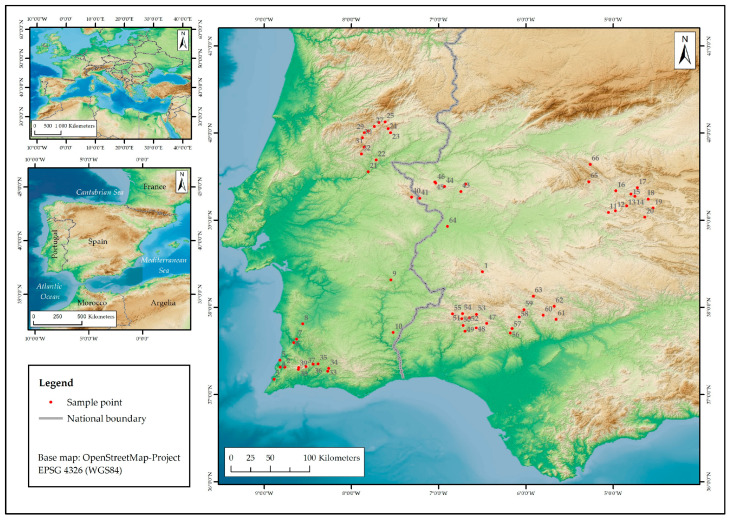
Distribution of *L. stoechas* subsp. *luisieri* sampling points utilized in this study.

**Figure 4 plants-14-03435-f004:**
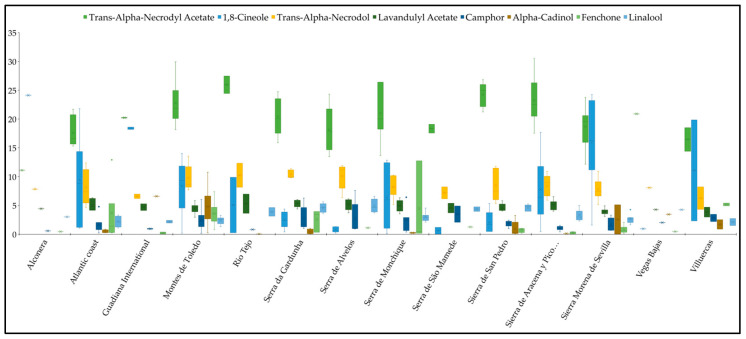
Box plot shows the distribution of the eight most prevalent compounds in the essential oil of *L. stoechas* subsp. *luisieri*, categorised according to zone.

**Figure 5 plants-14-03435-f005:**
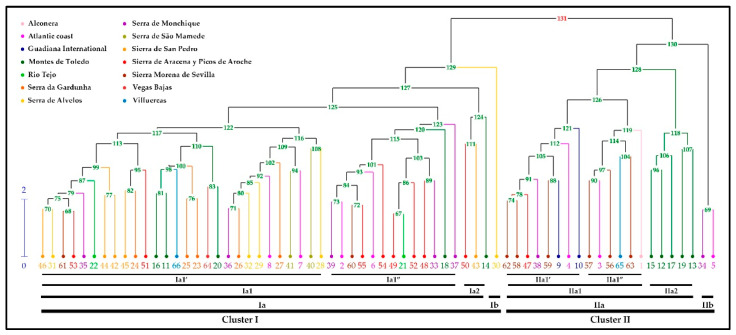
Hierarchical clustering of the percentage composition of the 66 samples of essential oils of *L. stoechas* subsp. *luisieri* analyzed (Note. Population number: 1–66).

**Figure 6 plants-14-03435-f006:**
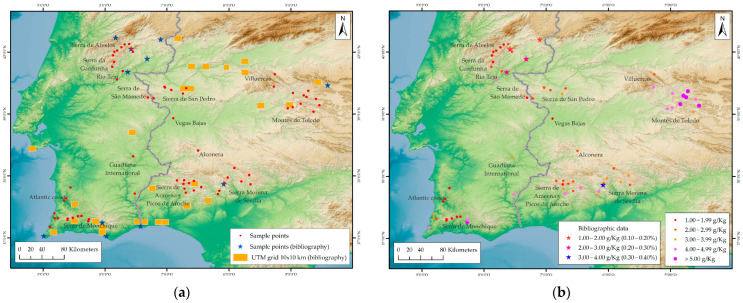
(**a**) Distribution map of sampling points for the analysis of the essential oil from *L. stoechas* subsp. *luisieri*. (**b**) Distribution map of essential oil yield data (stars—bibliography references and points—sampling points in this study).

**Figure 7 plants-14-03435-f007:**
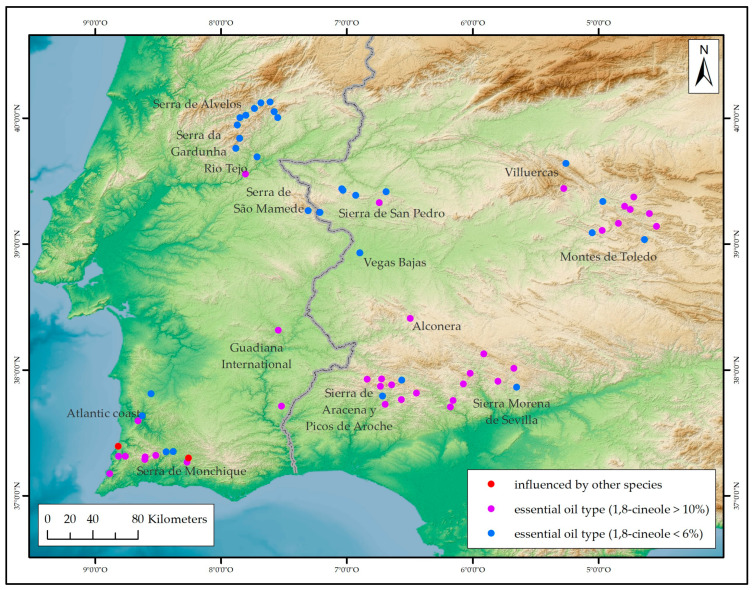
Geographical distribution of *L. stoechas* subsp. *luisieri* essential oil samples, based on the percentage of presence of 1,8-Cineole and other compounds indicative of the presence of other species or hybrids.

**Table 1 plants-14-03435-t001:** List of sampling areas for *L. stoechas* subsp. *luisieri*.

Zone	Type	Country	N	IDs	Elev.
Alconera	MR	ES	1	1	608
Atlantic coast	CZ	PT	7	2–8	36–112
Guadiana International	AP	PT	2	9–10	192–194
Montes de Toledo	MR	ES	10	11–20	515–824
Rio Tejo	AP	PT	2	21–22	204–373
Serra da Gardunha	MR	PT	5	23–27	425–617
Serra de Alvelos	MR	PT	5	28–32	311–829
Serra de Monchique	MR	PT	7	33–39	104–764
Serra de São Mamede	MR	PT-ES	2	40–41	362–526
Sierra de San Pedro	MR	ES	5	42–46	362–500
Sierra de Aracena y Picos de Aroche	MR	ES	9	47–55	260–709
Sierra Morena de Sevilla	MR	ES	8	56–63	296–709
Vegas Bajas	AP	ES	1	64	179
Villuercas	MR	ES	2	65–66	593–912

Note. N: number of populations collected per zone, IDs: identification number assigned to each population, Elev.: elevation (m. a. s. l.), Type (MR: mountain range, CZ: coastal zone; AP: alluvial plain), Country (ES: Spain, PT: Portugal).

**Table 2 plants-14-03435-t002:** Essential oil yield (g/kg) of *L. stoechas* subsp. *luisieri* for each zone.

Zone	N	Yield (g/kg)		
		µ ± σ	min.	max.
Alconera	1	2.65		
Atlantic coast	7	2.67 ± 1.24	1.42	4.43
Guadiana International	2	4.25 ± 1.07	3.46	4.97
Montes de Toledo	10	5.20 ± 1.29	3.01	8.17
Rio Tejo	2	3.31 ± 1.40	2.34	4.30
Serra da Gardunha	5	2.23 ± 0.60	1.42	2.95
Serra de Alvelos	5	1.46 ± 0.45	1.08	2.05
Serra de Monchique	7	2.62 ± 1.79	1.34	5.94
Serra de São Mamede	2	2.30 ± 0.47	1.97	2.64
Sierra de San Pedro	5	2.78 ± 0.46	2.20	3.27
Sierra de Aracena y Picos de Aroche	9	2.87 ± 0.87	1.79	4.25
Sierra Morena de Sevilla	8	3.11 ± 0.66	2.10	4.14
Vegas Bajas	1	1.75	1.75	
Villuercas	2	3.42 ± 0.82	2.80	4.03
	66	3.08 ± 1.42	1.077	8.17

Note. µ ± σ: plus or minus one standard deviation from the mean, min.: minimum value; max.: maximum value.

**Table 3 plants-14-03435-t003:** Major compounds are present in the essential oil of *L. stoechas* subsp. *luisieri*.

Components	µ ± σ	min.	max.	N	N1	N2	N3	N4	N5
*trans*-α-Necrodyl Acetate	20.68 ± 4.17	11.11	30.50	66	60	6	0	0	0
1,8-Cineole	7.79 ± 7.14	0.04	24.24	45	6	20	7	12	21
*trans*-α-Necrodol	8.66 ± 2.18	4.32	13.57	66	0	37	22	7	0
Lavandulyl Acetate	4.74 ± 0.91	2.95	6.96	66	0	1	22	43	0
Camphor	2.01 ± 1.55	0.16	7.61	30	0	1	3	26	36
α-Cadinol	2.24 ± 2.76	0.02	10.78	13	0	1	3	9	21
Fenchone	2.23 ± 2.91	0.01	12.92	17	0	0	6	11	31
Linalool	3.25 ± 1.19	1.09	6.58	62	0	0	3	59	3
5-Methylene-2,3,4,4-tetrame-2-Cyclopentenone	2.73 ± 0.70	1.26	4.13	58	0	0	0	58	8
Cymene Isomer	2.01 ± 0.79	0.09	3.29	50	0	0	0	50	15
Viridiflorol	1.96 ± 0.81	0.01	3.43	36	0	0	0	36	28
Unknown XIII	2.03 ± 0.86	0.14	3.71	30	0	0	0	30	24
*cis*-α-Necrodol	1.73 ± 0.27	1.10	2.29	29	0	0	0	29	37
α-Pinene	1.73 ± 0.98	0.41	5.36	20	0	0	0	20	46
Arbozol	1.69 ± 0.38	0.84	2.57	20	0	0	0	20	46
Unknown XVI	1.52 ± 0.70	0.09	2.89	16	0	0	0	16	46
Unknown XI	1.56 ± 0.40	0.57	2.31	13	0	0	0	13	43
Unknown I	1.60 ± 0.22	1.11	2.10	9	0	0	0	9	47
Eudesma-3,7(11)-diene	1.07 ± 0.59	0.12	3.24	6	0	0	0	6	60
Unknown X	1.80 ± 0.26	1.24	2.08	6	0	0	0	6	4
*cis*-α-Necrodyl Acetate	1.38 ± 0.28	0.82	1.92	5	0	0	0	5	61
Unknown XIV	1.38 ± 0.90	0.66	3.60	3	0	0	0	3	7
Copaborneol	1.01 ± 0.37	0.04	1.78	1	0	0	0	1	59
Unknown XX	0.80 ± 0.42	0.05	2.09	1	0	0	0	1	53

Note. µ ± σ: plus or minus one standard deviation from the mean, min.: minimum value; max.: maximum value, N: number of occurrences in the top ten, N1: number of times in first place, N2: number of times in second place, N3: number of times in third place, N4: number of times between fourth and tenth place, N5: number of times in chemical composition outside of top ten.

**Table 4 plants-14-03435-t004:** Range of values (in percentage) known compounds (which at least once are among the ten most abundant) in the essential oil of *L. stoechas* subsp. *luisieri*, grouped according to cluster analysis.

Components	Cluster I							Cluster II							
	Ia1′		Ia1″		Ia2		Ib	IIa1′		IIa1″		IIa2		IIb	
	min.	max.	min.	max.	min.	max.		min.	max.	min.	max.	min.	max.	min.	max.
α-Pinene	0.41	3.62	0.74	2.12	1.81	2.40	4.77	0.93	1.96	1.01	3.23	1.92	5.36	1.45	1.50
Cymene Isomer	0.09	3.29	0.17	3.16	0.26	1.37	1.91	1.85	2.76	1.80	2.36	a	2.31	2.04	2.06
1,8-Cineole	**0.04**	**5.63**	**5.38**	**12.84**	1.35	7.56	1.33	12.42	18.60	**19.84**	**24.24**	**10.50**	**14.01**	12.86	13.06
Fenchone	a	7.39	a	1.40	0.39	2.75	**a**	a	5.31	0.34	5.41	2.32	4.88	**12.76**	**12.92**
Camphor	0.38	6.27	0.64	6.42	0.16	1.90	**7.61**	0.23	2.90	0.58	3.45	1.36	6.00	1.56	2.72
Linalool	1.52	6.58	1.89	4.99	1.72	5.25	4.62	1.26	2.91	1.09	3.01	1.31	2.87	2.10	2.10
*trans*-α-Necrodyl Acetate	15.30	26.42	17.54	28.15	**26.83**	**30.50**	13.48	**16.59**	**20.46**	11.11	19.48	18.17	26.04	13.64	15.68
Lavandulyl Acetate	3.72	6.96	3.65	6.35	3.92	6.53	3.73	3.75	5.28	3.00	4.43	2.95	5.83	3.55	4.15
*cis*-α-Necrodyl Acetate	0.91	1.80	1.09	1.81	1.21	1.92	0.84	0.95	1.67	0.84	1.38	1.10	1.72	0.82	1.16
Arbozol	0.84	2.57	1.25	2.55	1.33	1.67		1.39	2.24	1.11	1.70	1.16	1.96	1.57	1.58
5-Methylene-2,3,4,4-tetrame-2-Cyclopentenone	1.57	4.00	1.59	3.51	1.26	2.56	3.04	1.89	4.13	1.83	3.15	1.33	1.77	2.23	2.23
*trans*-α-Necrodol	4.59	12.38	5.92	13.57	6.71	10.48	6.36	6.22	10.93	4.32	7.84	7.70	8.85	5.16	5.43
*cis*-α-Necrodol	1.58	2.29	1.34	2.22	1.32	1.88	1.10	1.36	1.96	1.27	1.69	1.27	1.55	1.46	1.46
Eudesma-3,7(11)-diene	0.20	2.29	0.12	2.69	0.14	0.99	3.24	0.53	1.39	0.40	1.31	0.40	2.01	1.42	1.76
Viridiflorol	0.44	3.43	a	3.09	0.14	2.50	2.96	0.56	2.73	1.12	2.07	a	0.92	1.44	1.45
Copaborneol	a	1.78	a	1.43	a	0.82	1.48	0.19	1.43	0.54	0.95	0.04	0.45	0.71	0.72
α-Cadinol	a	4.85	a	0.25	a	6.09	a	a	6.60	a	5.10	**2.99**	**10.78**	0.31	0.31

Note. min.: minimum value; max.: maximum value; a: compound not present. Bold: dominant compounds for each cluster.

## Data Availability

The original contributions presented in this study are included in the article. Further inquiries can be directed at the corresponding author.

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
