# Peer review of "Lavandula stoechas subsp. luisieri (Rozeira) Rozeira: Variability of Chemical Composition of Essential Oil in Wild Populations"

_plants, 2025, doi:10.3390/plants14223435_

Round 1
Reviewer 1 Report
Comments and Suggestions for Authors
Giving value to a territory's plant genetic resources should be a priority objective for all countries, as they contribute enormous value to the biodiversity of each enclave and thus allow for an increase in the value of biodiversity.
The work presented to Plants on the phytochemistry of Lavandula stoechas subsp. luisieri is highly relevant, as its chemical composition is studied across a large part of its distribution area and with a large number of wild populations studied (66). It is a well-written work with a well-conceived and executed methodology.
Below, I will make a series of comments or suggestions for the authors to improve the manuscript:
ABSTRACT
Include the relevance of necrodol derivatives. Mention the potential industrial interest of the results.
INTRODUCTION
A broad distribution is presented, clarifying relevant aspects of the taxon under study and related taxa. Relevant bibliography on the phytochemistry of the plant and its interesting industrial potential is also cited.
Although all the information is well cited, I would like to include the bibliography used to create the distribution maps, as they do not seem to coincide with the basic work on the flora of the Iberian Peninsula (Flora Iberica).
MATERIALS AND METHOD
The methodology is well presented and can be replicated in all cases.
It remains to be seen whether control material has been collected and included in a reference herbarium for each of the populations analyzed. This process is essential for replicating the study.
RESULTS
The results are presented correctly, highlighting the quality of the tables and figures. The statistical analysis performed is also noteworthy, offering quality results.
DISCUSSION
Although the discussion of the results is correct, I believe that several subsections should have been included, as has been done in the Results section. This would allow for a much better follow-up of the discussion.
CONCLUSION
The conclusion presented reads more like a summary of the discussion. I advise the authors to draft new conclusions that, for example, show the potential value of the results for the aromatic and medicinal plant industry.
Author Response
Comments 1: ABSTRACT
Include the relevance of necrodol derivatives. Mention the potential industrial interest of the results.
Response 1: Agree.
Comments 2: INTRODUCTION
A broad distribution is presented, clarifying relevant aspects of the taxon under study and related taxa. Relevant bibliography on the phytochemistry of the plant and its interesting industrial potential is also cited.
Although all the information is well cited, I would like to include the bibliography used to create the distribution maps, as they do not seem to coincide with the basic work on the flora of the Iberian Peninsula (Flora Iberica).
Response 2: Thank you for pointing this out. The distribution maps have been compiled from data obtained from a thorough review of herbarium specimens (rather than bibliographic references). In order to achieve this objective, a range of websites were consulted, and a number of herbariums were approached with a view to obtaining high-resolution scanned images of the subspecies L. stoechas subsp. stoechas and L. stoechas subsp. luisieri. A total of 1,039 herbarium sheets from 27 herbaria (BC, BG, BM, BR, CHSC, COI, DOV, E, FI, G-GDC, GJO, HSS, K, LI, LISU, MA, MW, NY, OSC, P, PI, PRC, RAB, RM, USCS, VAL, W) were reviewed.
Following the characterisation and identification of the relevant materials, a comprehensive analysis of their potential distribution has been conducted. This analysis is founded upon climatic and topographical data, which has been meticulously collected and processed.
This work on the morphological diversity and distribution of the subspecies of L. stoechas is currently unpublished, and we believe it is not appropriate to include it in this manuscript. Its inclusion would extend the length of the manuscript, and no direct relationship has been observed between morphological characteristics and the chemical composition of the essential oil.
Added to the text of Figure 2 (Source: Study of herbarium vouchers. Unpublished work) to indicate the unpublished nature of this work.
On the other hand, the reviewer indicates that the distribution shown does not match Flora Iberica [reference number 15]. This assertion is particularly salient in the context of L. stoechas subsp. stoechas. The distribution exhibited by this taxon is more consistent with that demonstrated by Upson & Andrew in The Genus Lavandula [reference number 1], as referenced in the text pertaining to the distribution of this subspecies (lines 64-67). In L. stoechas subsp. luisieri, the distribution is consistent with that documented in Flora Iberica [15], Upson & Andrew [1] and the research on section stoechas conducted by Vázquez et al. for the south-west of the Iberian Peninsula [reference number 16], as outlined in the text (lines 68-73).
Comments 3: MATERIALS AND METHOD
The methodology is well presented and can be replicated in all cases.
It remains to be seen whether control material has been collected and included in a reference herbarium for each of the populations analyzed. This process is essential for replicating the study
Response 3: Agree. A paragraph with that information has been added.
Comments 4: RESULTS
The results are presented correctly, highlighting the quality of the tables and figures. The statistical analysis performed is also noteworthy, offering quality results.
Response 4: Agree.
Comments 5: DISCUSSION
Although the discussion of the results is correct, I believe that several subsections should have been included, as has been done in the Results section. This would allow for a much better follow-up of the discussion
Response 5: Agree. Four sections have been added to the discussion:
3.1. Geographical Distribution
3.2. Essential Oil Yield
3.3. Chemical Composition of Essential Oil
3.4. Principal Compounds of Essential Oil.
Comments 6: CONCLUSION
The conclusion presented reads more like a summary of the discussion. I advise the authors to draft new conclusions that, for example, show the potential value of the results for the aromatic and medicinal plant industry.
Response 6: Thank you for pointing this out. The conclusions have been modified to include information on the potential value of this subspecies as an aromatic and medicinal plant for industry.

Reviewer 2 Report
Comments and Suggestions for Authors
Review report for plants-3958701
The study of Marquez-Garcia et al. deals with the essential oils of an endemic of the Iberian Peninsula taxon, Lavandula stoechas subsp. luisieri. Although this is not the first time this taxon’s essential oils have been studied, the work of Marquez-Garcia et al. provides many new data regarding the geographical diversity almost throughout its distribution area. A sufficient number of populations were studied and the sampling design and the analytical and statistical methods used were appropriate and sufficient. The discussion of the results in relation to the existing bibliographic information was detailed and informative. The conclusions are consistent with the evidence and arguments presented. Overall, the work is well-designed and well-written.
I believe that it is more correct not to use the term chemotype, but the term essential oil type instead. This is because a chemotype is a distinct chemical race that should be distinguished based on clear qualitative characteristics. In the case of the populations you are examining, the differences of the essential oils are relatively small and mainly only quantitative.
I suggest to add a figure concerning the distribution of the two chemotypes, similar to Figure 6, to depict the existence of any geographical variation.
Author Response
Comments 1: I believe that it is more correct not to use the term chemotype, but the term essential oil type instead. This is because a chemotype is a distinct chemical race that should be distinguished based on clear qualitative characteristics. In the case of the populations you are examining, the differences of the essential oils are relatively small and mainly only quantitative.
Response 1: Agree. The manuscript has been modified in accordance with the reviewer's suggestions.
Comments 2: I suggest to add a figure concerning the distribution of the two chemotypes, similar to Figure 6, to depict the existence of any geographical variation.
Response 2: Agree. Figure 7 has been included in the discussion section to illustrate the distribution of essential oil types.

Reviewer 3 Report
Comments and Suggestions for Authors
Please see attached document.

Author Response
Comments 1: Section 1. Introduction
- Line 68 reads “...at altitudes below 900-950 m a. s. l.” This abbreviation is not defined as meters above sea level until line 120 in the Results section. Please define the first occurrence of use.
Response 1: Agree. The manuscript has been modified in accordance with the reviewer's comments.
Comments 2: Section 2. Results
2.2 Chemical Composition
Nice job clarifying that compounds were detected - I often see authors state that a sample contained a certain number of compounds when in reality we only know what we can detect as the authors have correctly stated.
Table 3 in the text and Supplemental S2a-d list “unknown sesquiterpenols” and “unknown esters” and “unknown Mw=153” to list a few. Please include some verbiage to explain how these unknown compounds have been determined to be sesquiterpenols and how the molecular weight was determined for the indicated unknown compounds. If the mass spectrum has a base peak at 153 Da, that is not sufficient evidence to determine the unknown has that molecular weight.
Response 2: Thank you for pointing this out. The molecular weight was determined by utilizing the base peak obtained from the mass spectrum. It is important to note that this does not guarantee absolute accuracy of the value 100%, as the reviewer correctly observes.
Conversely, the estimation of molecular structure for indeterminate compounds, such as sesquiterpenes and esters, has been facilitated through the calculation of molecular weight values utilizing the base peak. Subsequently, based on the molecular weight, the presence of oxygen atoms that may be present in the compound has been estimated. This does not provide absolute certainty; rather, it suggests a high probability of correctness.
As demonstrated by the data presented in the study, Lavandula stoechas subsp. luisieri is one of the most complex oils in existence, containing numerous compounds that are not yet available even in the most comprehensive libraries.
For all these reasons, we have removed all references to molecular weight or chemical structure of the undetermined compounds. This will prevent any possible confusion regarding the chemical nature of these compounds in future studies on the chemical composition of the essential oil of this species.
Comments 3: Section 3. Discussion
Line 311 it appears that punctuation is missing and the sentence is awkward “The most recent populations the focus of the study in Portugal…” Should there be a period after “...period [54,59]”?
Lines 321-322 states “The observed percentage variation in the compounds that constitute the essential oil of L. stoechas subsp. luisieri is largely consistent with the results obtained in this study.” To which study or studies does this statement refer? Please clarify.
Response 3: Agree. Line 311 has been corrected (period included after [54,59]) and the reference to Lines 321-322 has been included.
Comments 4: Section 4. Materials and Methods
4.2. Collection of Plant Materials
- Please explain how the samples were identified to ensure the correct plant was harvested in each location.
- Were voucher samples retained? If so, please indicate the location of voucher samples in text.
4.5. Statistical Analysis
- Line 396 - please replace “Nueva” with “New”.
Response 4: Agree. 4.2. Collection of Plant Materials. Agree. A paragraph with the explanation has been included.
4.5. Statistical Analysis. Line 396 has been corrected.
Comments 5: Section 5. Conclusions
The last sentence of Section 1. (Introduction) states that “The objective of the present study is twofold: firstly, to ascertain the presence or absence of diverse chemotypes in L. stoechas subsp. luisieri, and secondly, to identify the chemical components that contribute to its economic value.” The authors have clearly presented evidence to support the occurrence of diverse chemotypes in L. stoechas susp. luisieri, but the support of the economic value is lacking in the Conclusions. Please mention the economic value of the chemotypes (e.g., anti-inflammatory, antioxidant, antifeedant, etc.) or alter the statement at the end of Section 1.
Response 5: Thank you for pointing this out. We have modified the conclusions to indicate the economic value of Lavandula stoechas subsp. luisieri.

Round 2
Reviewer 1 Report
Comments and Suggestions for Authors
The authors of manuscript 3958701 in Plants have taken into account all of my suggestions and have addressed some of my questions regarding certain aspects of the work.
Among the changes made, I would highlight the discussion sections and the new conclusions. They have also confirmed the existence of reference material in a reference herbarium.
In the Introduction section, the distribution of the plant under study has been clearly explained, indicating that it is based on a work that has not yet been published. Given this, I propose that this information not be presented and that the maps be removed from the work. However, I leave this proposed modification to the authors themselves.
After a final reading of the work, I consider it suitable for publication.